# Suitability of Banana and Plantain Fruits in Modulating Neurodegenerative Diseases: Implicating the In Vitro and In Vivo Evidence from Neuroactive Narratives of Constituent Biomolecules

**DOI:** 10.3390/foods11152263

**Published:** 2022-07-29

**Authors:** Barnabas Oluwatomide Oyeyinka, Anthony Jide Afolayan

**Affiliations:** Medicinal Plants and Economic Development (MPED) Research Centre, Botany Department, University of Fort Hare, Alice 5700, South Africa; barnabastom@yahoo.com

**Keywords:** anti-inflammation, biomechanism, biomolecules, fruits, *Musa* species, neurodegenerative diseases, neuroactivity

## Abstract

Active principles in plant-based foods, especially staple fruits, such as bananas and plantains, possess inter-related anti-inflammatory, anti-apoptotic, antioxidative, and neuromodulatory activities. Neurodegenerative diseases affect the functionality of the central and peripheral nervous system, with attendant cognitive deficits being hallmarks of these conditions. The dietary constitution of a wide range of bioactive compounds identified in this review further iterates the significance of the banana and plantain in compromising, halting, or preventing the pathological mechanisms of neurological disorders. The neuroprotective mechanisms of these biomolecules have been identified by using protein expression regulation and specific gene/pathway targeting, such as the nuclear and tumor necrosis factors, extracellular signal-regulated and mitogen-activated protein kinases, activator protein-1, and the glial fibrillary acidic protein. This review establishes the potential double-edged neuro-pharmacological fingerprints of banana and plantain fruits in their traditionally consumed pulp and less utilized peel component for human nutrition.

## 1. Introduction

Plant-based foods, such as fruits, contain natural active principles, ranging from primary metabolites (nutritive factors, vitamins, and minerals) to secondary metabolites (phytochemicals) [1]. These bioactive principles play significant roles in mitigating several chronic diseases [2,3,4,5]. Some epidemiological studies have identified antioxidant-rich secondary metabolites, such as flavonoids and anthocyanin, for their anti-inflammatory, antiproliferative, and ameliorative roles in neurological disorders [6,7]. Generally, natural polyphenols express their neuroprotective capacity by relying on their mechanism ability to cross the blood–brain barrier to scavenge the pathological concentrations of reactive oxygen and nitrogen species [8]. Equally, polyphenols modulate a series of mediating cell-signaling pathways of pathological diseases [9].

## 2. Overview and Prevalence of Neurodegenerative Diseases

Neurodegenerative diseases essentially relate to any pathological condition that primarily affects the neuron [10]. They affect the central nervous and are typified by the regression and progressive decline of neurological functioning and cognitive deficit [11], which results in major conditions, such as Alzheimer’s, Parkinson’s, and dementia [12,13]. This challenge is particularly devastating in aging populations, with Alzheimer’s disease affecting about 40 million people globally [14,15]. Exposure to multiple factors (environmental and genetic) contributes to the onset of neurodegenerative diseases. Neurotoxic metal pollutants, such as mercury, lead, cadmium, and arsenic, have been identified with Alzheimer’s and Parkinson’s disease, oxidative stress, neuronal death, mitochondrial dysfunction, modulation of metal homeostasis, and aggregation of α-synuclein proteins [16,17,18]. A schematic description of the aforementioned is depicted in Figure 1. It is even more concerning that these environmental factors can cause damage to the neurologic system via epigenetic mechanisms and then trigger neurodegenerative disease in later years [18,19]. Neurodegeneration consists of a series of pathways that have been closely linked to its inflammatory process and, in particular, the pro-inflammatory cytokines implicated in the pathogenesis of functional and neurologic impairment [20].

The burden of neurological disorders and conditions has necessitated reliable data to enhance effective health planning approaches. Epidemiological data for neuro conditions, particularly Parkinson’s, dementia, and amyotrophic lateral sclerosis, have been reported in the last two decades [21,22,23,24,25]. Frontotemporal dementia is typically identified in middle age, with reports of about 13% occurrence in people below the age of 50 [26]. Furthermore, Ref. [26] a systematic analysis of a demographics-based study was carried out and estimated about 2–31 frontotemporal dementia incidences per 100,000 people in Europe and developed countries of Asia and North America [26]. An average occurrence of Multiple System Atrophy (MSA) (3 per 100,000 people), average onset age (54–61 years), and demographic prevalence in Europe and North America have been reported in various studies [27,28,29].

## 3. Fruits in Neurodegenerative Prevention/Management

The brain is especially a centrally significant organ of the nervous system which requires a healthy diet, with fruits potentially offering a wide range of rich nutrient supplies [30]. Fruits represent a major dietary component across Western and Asian demographics. They have been reported to have positive synergy with and relation to chronic disease management [31,32,33,34]. Examples of these common fruits include *Malus domestica, Persea americana, Musa sinensis, Citrus limon, Musa paradisiaca, Pyrus communis, Citrus sinensis, Fragaria ananassa,* and *Ananas comosus* (Figure 2).

Date palm fruits (*Phoenix dactylifera*) have been reported in studies for their potential biological capacity in nephroprotective, hepatoprotective, and anticancer activity [35,36,37]. Recently, the inhibitory effect of avocado juice against trypsin aggregation has been reported: a formation process associated with several neurological diseases [38]. Similarly, other reports identified the beneficial role of berries (blackberry and blueberry) with regards to the obstruction of the central nervous system and cognitive deficit [39,40]. The cactus pear (*Opuntia ficus-indica*) fruit has also been reported to have the biological capacity to modulate neuron excitation in a distributive manner across sectors of the brain [41].

Over time, a host of bioactive molecules cutting across nutritive and antinutritive factors, vitamins, minerals, and secondary metabolites have been screened, evaluated, and elucidated in several species and varieties of banana and plantain fruits, as shown in Table 1.

## 4. Neuroprotective Mechanistic Narratives of Active Principles and Mineral Elements Listed in Banana and Plantain fruits

### 4.1. Tannins

Natural or dietary tannins are nutraceutical factors that have been reported in cognitive impairment amelioration [127,128]. Furthermore, dose-dependent cell-line neuroprotective activity has been reported in natural gallotannins and ellagitannins against H_2_O_2_-based oxidative damage, using the hybridoma cell line NG108–15 [129]. Similarly, gallotannin derivatives increased the cell viability of the hybrid neuroblastoma–glioma. Ellagitannin derivatives of *Phyllagathis rotundifolia* were identified for neuroprotective capacity by dose-dependent neuronal cell protection at concentration levels between 6.25 and 100 µM [129].

In other in vivo model studies, the neuromodulatory mechanisms of tannic acid supplementation were reported against brain injury [49,130]. This observation is a potential neurotherapeutic management avenue, with regards to issues such as neuron dysfunction and behavior alteration. Notably, tannic acid supplementation deploys the PGC-1α/Nrf-2/Ho-1 molecular activation mechanism that modulates neurological maladies and factors such as brain edema, pro-inflammatory cytokine expression, and the glial fibrillary protein immunoreactivity [49]. The neuroprotective activity of tannic acid has been elucidated in a study identifying its anti-inflammatory tendencies, with regards to the occlusion of the mid-cerebral artery in vivo [48]. The same study equally indicated tannic acid suppression of neuronal loss and downregulation of GFAP expression, leading to the protection against brain damage [48]. In vitro and in vivo studies have identified condensed tannin (proanthocyanidin) for its mechanisms of action, such as neuron loss attenuation and exhibition of anti-CHE activity, which offer neuroprotective roles against Parkinson’s and Alzheimer’s diseases, respectively [131,132].

### 4.2. Phenolic Acid

Phenol acid is a major polyphenolic member found in fruits and grains. A study has recently pointed out that a phenol-rich diet lowers the dispositional risk of depression [133].

### 4.3. Quercetin

Quercetin is another compound of flavonoid polyphenolic extraction, with fruits, vegetables, and herbs being the chief bio-stores. In vivo experiments have identified the role of quercetin in nerve regeneration and apoptotic index reduction, including axonal structure healing [134]. Another study has reported the neuroprotective properties of quercetin supplementation through the inhibition of aluminum-induced oxidative damage by halting aluminum buildup in the hippocampus and striatum sectors of the brain [135]. Furthermore, quercetin significantly improved muscle coordination and cognition in vivo [135]. Quercetin has a potential capacity to inhibit mitochondrial dysfunction/oxidative stress induced by oxaliplatin. Moreover, 10 and 20 mg/kg quercetin concentrations inhibited focal cerebral ischemic cell apoptosis by mechanistically activating the BDNF-TrkB-PI3K/Akt signal pathway [58,136]. Equally reported is the fact that quercetin shrunk the volume of infarct in the damaged contralateral hemispheric region of the brain in an in vivo model, from an initial 36.2 ± 4.8% to 24.8 ± 2.7% and 21.7 ± 3.2% upon 10 and 20 mg/kg daily treatments, respectively [58].

In a recent study, the modulatory effect of quercetin (30 mg/kg dosage) against lipopolysaccharide (neuroinflammatory trigger factor) was reported [137]. This was expressed by rescuing neuronal degeneration; mitochondrial apoptotic mechanisms, such as Bax/Bcl2, PARP-1 cleavage; and the stemming of caspase-3 activity in the cortico-hippocampal locations of the brain.

### 4.4. Rutin

Rutin is a typical flavonoid compound with abundance in plants. Studies have demonstrated a wide spectrum of rutin activities, such as antioxidant, cytoprotective, and neuroprotective capacities [63,138,139,140,141]. Another study identified rutin suppression of proinflammatory cytokine activity through the mechanistic shrinkage of microglia-situated production of TNF-α and IL-1β [62]. The ameliorator role of rutin against neurodegeneration was reported in neuroblastoma cell-based and in vivo models, where neurodetrimental factors such as apoptosis, episodic deficit in memory, and reactive oxygen species were attenuated [142]. Research findings have indicated that rutin pretreatment at 25 mg/kg daily dosage significantly ameliorated neuroinflammatory mechanisms by depleting the expressive activities of poly ADP-ribosyl polymerase, glutathione reductase, and glutathione peroxidase in the hippocampus [63].

In the same vein, recently, the neuroprotective capacity of rutin, expressed in its anti-inflammatory activity, has also been evaluated [67]. In this in vivo study, 30 mg/kg rutin was reported to have inhibited the p38 MAPK pathway that influences inflammation, thereby ameliorating injury to the spine. The PRP (106–126) model system was used to evaluate neurotoxicity in the hippocampus cell line (HT22), with the rutin treatment (5–50 µg/mL) identified in blocking the generation of reactive oxygen species and caspase-3 activity, a hallmark of the toxic effects of the PRP (106–126) [64].

Furthermore, the modulatory action of rutin treatment (40 to 80 mg/kg) over nitric oxide has been implicated as a possible link to its neuroprotective mechanisms, particularly in head-trauma-related cognitive dysfunction [143].

In vitro and in vivo neuroprotective evidential narratives show that rutin (100 mg/kg dosage) depleted the levels of neurotoxic Aβ oligomer and interleukin levels in the brain of Alzheimer-model mice [65]. More in vitro reports have put forward the putative roles of rutin in depleting the formation and accumulation of Aβ 25–35 fibril and modulating the aggregation of Aβ, TNF-α, and interleukin-1 [61,62].

### 4.5. Carbohydrates

These are major nutritive factors that are obtainable in fruit, vegetable, grain, and cereal diets. An in vitro cell line study has identified the protective mechanism of oligosaccharides (50 µM) in the cells of neuroblastoma [73]. These oligosaccharide-treated cells were identified with proteins involved in neuroprotective biochemical mechanisms (TrKA receptor interaction and ERK1/2 pathway activation), as well as the inhibition of the MPTP expression that stimulates Parkinson’s disease [73]. Natural polysaccharides from *Lycium barbarum* fruits have also been reported to suppress p-JNK and p-ERK (a biomechanism that translates into neuromodulatory effects against neuron cell death upon hourly polysaccharide treatment at dosage 100–500 µg/mL) [72].

### 4.6. Lipids

Dietary lipids (fatty acids) are nutritive factors and are regulatory biomolecules for intracellular signaling, homeostasis, and functionality of the central nervous system [144].

In an in vitro neuronal stem cell study, it was also reported that the potential anti-stroke therapeutic option is the Omega-3 fatty acid (docosahexaenoic acid (DHA) [145]. The dietary fatty acid has been implicated in the preventive or attenuative roles in neuro disorder and degeneration [145]. Fatty acid (10^−9^ to 10^−8^ M) in this study significantly improved regenerative neurogenesis, partly due to its antioxidant properties. Even more pointedly, correlations have been drawn between increased fatty acid levels and improvement in brain structure markers, language, and proper cognitive levels [144,146]. Further in vitro evidence indicates the role of lipids in the increment of hippocampal neurite length at 1.5 µM supplemental dosage [146].

### 4.7. Magnesium

Magnesium is a nutritive mineral component of diets that has a wide range of significance to health. It is a key driver of nerve transmission in the nervous system and also inhibits hyper-excitatory-induced cell death and epileptogenesis [147,148]. Neuromechanistically, magnesium interacts with the N-methyl-D-aspartate receptor [78]. Furthermore, a couple of studies have implicated magnesium pretreatment in neuron protection [149,150].

### 4.8. Zinc

Zinc is an important mineral element that plays a significant role in nutrition and health, including regular brain function. An in vitro study has identified neurological mechanisms of zinc, such as MBP protein upregulation, cognitive impairment amelioration, and modulatory effect on the regenerative sprouting of the mossy fiber of the hippocampus [81]. Consequently, zinc supplementation (246 mg/kg) presents a therapeutic mechanism in compensating and repair of the damage to neuron membrane in developmental seizures [81]. Zinc deficiency has been examined and identified with impairment of the proliferation of neuron precursors, caspase, and apoptotic inducement [80,151].

### 4.9. Copper

Copper is a trace element of nutritive value and is an enzymatic structural component. It is involved in physiological processes, such as metabolic regulation and hormonal biosynthesis [152]. Deficiency in copper levels induces the depletion of several enzyme activities, with precursory implications in the Menkes neurological disorder [112]. For instance, dopamine-β-hydroxylase (DBH) enzyme is dependent on copper for its neurotransmission role in translating dopamine to norepinephrine [112]. Abnormal neurochemical trends in Menkes disease result when the copper-dependent DBH enzyme activity is altered [153,154]. In vitro copper-related therapeutic experimental propositions have been reported through the CuSO_4_ pretreatment (10 µmol/kg) modulation of Parkinson’s-related protein nitration and depletion of dopamine [155].

### 4.10. Alkaloid

Alkaloids are non-nutritive phytochemical compounds with neuroactive potentials. Plant-derived alkaloid compounds (30-day 1 g/kg dosage) have been experimented with, in an in vivo model [156]. Resultantly, modulatory effects were observed against Aβ peptide accumulation and plaque buildup, as well as cognitive deficit reversal. In earlier work, p53-mediated alkaloid-based compounds have been reported for activity in vivo against neurotoxicity at 200 mg/kg [83]. Alkaloid-based compounds have the molecular capacity to upregulate the activation of synaptic plasticity genes, such as BDNF, MAP2, GAP43, PSD-95, and KLK8 [83,84]. Similarly, plant-based alkaloids from sources such as *Clausena lenis* and *Isatis indigota* have been reported for their in vitro neuroprotective activity against cell death in SH-SY5Y human neuroblastoma cells, at (0.68 to 18.76 µM) and (25 to 100 µM), respectively [157,158].

### 4.11. Saponin

Saponins are bioactive compounds found in plant-based foods. Preclinical studies have reportedly implicated saponins in neuromodulatory capacity and memory enhancement potential. The long-term effect of bacoside saponin (200 mg/kg) has been evaluated in vivo, with significant neuroprotection against aging-related cognitive decline [159]. Triterpenoid saponins of *Platycodi radix* have demonstrated inhibitory activity against neurotoxicity in cortical cells at 0.1 to 10 µM [160]. The neuroprotective capacity of red-ginseng-derived saponins (50–150 mg/kg) has been demonstrated in terms of significantly lowering malondialdehyde levels, while increasing catalase, superoxide dismutase, and glutathione levels [81]. Saponin (2 mg/kg/d) from *Radix trichosanthis* exhibited neuroprotective activity, as evidenced by less damage to neuron cells, and inhibition of the expression of p53 and p38 in subarachnoid hemorrhage [87]. Furthermore, the saponin neuroactivity could be a result of the p38 and p53 signal pathway mediation [87]. Isolated saponin from *Astragalus glycyphylloides* showed the most significant neuroprotective capacity, at 60 µg/mL, against oxidative stress in isolated brain synaptosomes [161].

### 4.12. Phytate

Phytic acid is a predominantly plant-based compound that tends to interact with the absorption or bioavailability of mineral nutrients. However, phytate possesses bioactive properties that were identified in a study, with implicative roles in potentially intercepting the buildup of Aβ in the brain and neuroblastoma cells [162]. The inhibition of β-secretase 1 and γ-secretase enzymes offers a potential biomechanistic pathway or trajectory in the aversion of Alzheimer’s disease. In an earlier finding, phytic acid was reported for inhibitory activity against amyloid-β-peptide pathogenesis in MC65 cell and in vivo model Tg2576 [89].

### 4.13. Vitamin B

Vitamin B is a water-soluble group of vitamins with subtypes [163]. They are essentially involved in red-blood-cell synthesis and metabolism and are sourced in foods such as fruits, vegetables, and grains. The B vitamins are co-factors of enzymes in a series of biochemical pathways [164].

Thiamine (vitamin B1) is essential in glucose metabolism; nerve membrane functionality; and myelin synthesis, including neurotransmitters such as serotonin and acetylcholine [165].

Nicotinamide (Vitamin B_3_), a precursor of NADH and NADPH coenzymes, has neuroprotective properties at low concentrations [166]. Similarly, a low dose of vitamin B_3_ (10 mM) significantly induced embryonic-stem-cell differentiation into neurons [167]. In addition, another report has implicated nicotinamide (0.1/1.0 mM) in the increased survival rate of PC12 cells, using a simulated model of Parkinson’s disease [168].

Pyridoxine (Vitamin B_6_), at 600 mg/kg, improved locomotor behavioral performance and ameliorated a cortical injury model in vivo [169]. Pyridoxine has also been implicated in the regulation of GABA levels, an inhibitory neurotransmitter linked to excitatory epileptic seizures [170].

Cobalamin (Vitamin B_12_) contributes to the protection of nerve fibers through myelin sheath synthesis [171]. In a recent study, vitamin B_12_ was experimentally demonstrated to have modulatory neuroprotective and anti-inflammatory activities against pneumococcal meningitis and hippocampus damage [172]. In addition, hippocampus apoptosis, triggered by bacterial meningitis, was diminished by vitamin B_12_ treatment (10 µL intramuscular dosage) [172].

### 4.14. Vitamin C (Ascorbic Acid)

Vitamin C is a very prominent and essential water-soluble vitamin member that is a rich component of food, particularly citrus fruits, berries, and vegetables. Evidence exists on the neuroprotective potential of vitamin C through the mechanism of increased generation of DOPA upon neuroblastoma cell line incubation in ascorbic acid (dosage 100 mM to 500 mM) [173]. Furthermore, ascorbic acid has been identified as a potential early phase treatment of Parkinson’s disease through the biomechanism of stimulating tyrosine gene expression at a specific dosage (200 µM) [173]. It also inhibited beta-amyloid apoptosis in neuroblastoma cell SH-SY5Y at 50 µM [174].

### 4.15. Vitamin E (Tocopherol)

These are fat-soluble compounds that are involved in key physiological frontiers, such as regulatory roles in gene expression and immune function. It has also been demonstrated that vitamin E exerted neuroprotective activity against MPP^+^-induced toxicity in SH-SY5Y cells and in vivo model of Parkinson’s disease, by utilizing the signaling pathways ERB-PI3K/Akt therein [96]. In other Parkinson’s studies, vitamin E (10^−3^ M) has been reported to block levodopa toxicity in neuroblastoma cell NB69, while in an in vivo BALB/c mice model, MPTP neurotoxicity was inhibited at 100 mg/kg ascorbic acid [175,176].

### 4.16. Selenium

Selenium is a trace mineral element that is usually required by the body in small quantities. It is a constituent of antioxidant-natured enzymes and selenoproteins. Selenium is found in plant foods, such as staples and cereals. The neuroprotective activity of selenium (1.5 mg/kg), along with N-acetylcysteine (NAC) in brain trauma apoptosis and oxidative stress, in the hippocampus has been reported [162].

Furthermore, organo-selenium compounds had inhibitory activity against Aβ peptide-induced toxicity and equally enhanced the levels of the nerve terminal marker SNAP-25 [101]. More recently, the neuroprotective activity of selenium-rich polysaccharides from *Rosa laevigata* fruit has been demonstrated against oxidative stress in neuroblastoma cells, particularly at a concentration of 100 µg/mL [157].

### 4.17. Phytosterol

Phytosterol has neuroactivity against cognitive deficits linked to high cholesterol levels. Phytosterol supplementation activity against neurological inflammation was evaluated in vivo [104]. Furthermore, phytosterol enhanced cholinergic function in the cerebral cortex through the restoration and depletion respectively [104]. Plant sterols have been hypothesized to possess the biomechanism of crossing the blood–brain barrier. They activate anti-inflammatory mechanisms that are involved in the buildup of amyloid-β clearance and depletion of β-secretase activity [105]. In another recent study, phytosterols were implicated neuroactive-wise in reducing Aβ plaque formation, as well as memory deficit amelioration [177].

### 4.18. Terpenoid

This is a large group of organic compounds (secondary metabolites) that are widely found in plants and have aromatic properties. Monoterpene (thymol) was identified for neuroprotective activity in an in vivo Parkinson’s disease model against dopamine-related neurodegeneration [178]. In the investigation, thymol, at a dose of 2.5 mg/kg for four weeks, attenuated neuronal loss and inflammation and was related to the preservation of antioxidant networks of defense. Fractions of *Hygrophila auriculata*–derived terpenoids (100 and 200 mg/kg) reflected neuroprotective activity in CA1 neurons of the hippocampus, hand in hand with antioxidative restoration [179]. Similarly, terpenoids have been marked for potential mechanisms, such as boosting the activity of glutamate decarboxylase enzyme, as well as GABA in the brain [107,180].

### 4.19. Glycosides

Glycoside has been reported in the neuroprotective frontier, as evidenced in the evaluation study against neuroinflammation and brain injury [103]. It biomechanically inhibited nuclear factor (NF-kB) and STAT3 activation in an experimental model of stroke.

In another narrative, phytoglycoside treatment (250 mg/kg) was identified to play recovery roles in the level of neurotransmitter hormones in the brain and serum in vivo [181].

### 4.20. Anthraquinones

Anthraquinone is the largest member of the quinones. It is a secondary metabolite with medicinal value. They are therapeutically relevant in central nervous system conditions, tumor and brain injury, cerebral hemorrhage, ischemic stroke, and Alzheimer’s disease [182]. Anthraquinones reduce and inhibit inflammation and apoptosis respectively [183,184]. In addition, they have been identified in therapeutic roles targeted at glutamate-related excitatory toxicity in brain trauma [185,186]. Other neuroprotective mechanisms have been identified in the same regard, such as inhibition of the ERK/MMP-9 pathway and amelioration of brain edema through the combined biomechanism of ERK/MMP-9 inhibition, as well as the arrest of the activation of the NADPH oxidase subunit, gp91phox [110]. Emodin anthraquinone (20 to 40 µM) has also been identified for its neuroprotective attenuation of TGF-β1 by inhibiting the NF-Kb pathway [187].

### 4.21. Anthocyanins

Anthocyanins are important phytochemicals found in fruits, flowers, and vegetables, to which they confer purple to red pigmentation. An in vivo model reported anthocyanin’s capacity to stem memory-breach condition at 200 mg/kg weekly dosage [188]. Equally, anthocyanin extracted from *Glycine max* was involved in the inactivation mechanism of ASK1-JNK/p38, thus modulating oxidative stress. Similarly, anthocyanin inhibited cytotoxicity in the SK-N-SH cell at a concentration level of 25 µg/mL [92]. Notably, studies evidenced that anthocyanin-rich fraction or extract treatment significantly stems the tide against interleukin-1β, tumor necrosis factor (TNF-α), and nitric oxide buildup [93,94].

### 4.22. Arginine

Arginine proffers therapeutic potentials in trauma-related brain injury [189]. It is, for instance, a precursor to the important brain and muscle tissue energy source, creatinine [189]. There have been reports of highly effective neuroprotective activity at 300 mg/kg L-arginine dosage in post-traumatic brain injury [190,191]. Furthermore, the focal arginase inhibitor (NW-hydroxy-nor-arginine) is also implicated in the in vivo amelioration of post-traumatic brain injury contusion [192,193]. Polyarginine peptide complexes have also been identified to be stroke neuroprotective in both in vitro and in vivo investigations [194,195,196]. In ischemic cerebral injury, arginine suppressed the HIF-1α/LDHA biomechanism that usually mediates inflammation in the microglia [112].

### 4.23. β-Carotene

This molecule is sourced from a wide range of fruits, such as bananas, mango, and apricots; and vegetables, such as spinach, pumpkin, and tomato. β-carotene treatment (10 to 50 mg/kg) improved cognitive output and neural functionality [81]. Further neuroprotective activity was reported in form of brain edema reduction in trauma. Then modulation of the Nrf2/Keap1 pathway was implicated in β-carotene’s alleviation of oxidative stress [81].

### 4.24. Lycopene

Dietary lycopene is widely found in vegetables and fruits. Lycopene, in an Alzheimer’s disease model, attenuated oxidative damage of the mitochondria and inhibited the activity of NF-kB, while equally regulating the proinflammatory cytokine expression [113,197]. This triggered suppression of Aβ buildup. Lycopene supplements have been reported in an in vivo Alzheimer’s disease study to improve cognitive factors via the mechanisms of lowering malondialdehyde level, marked attenuation of tau hyperphosphorylation, and enhanced activity of glutathione peroxidase (GSH-Px) [114].

### 4.25. β-Cryptoxanthin

Human cell line studies have reported beta-cryptoxanthin neuroactivity against hydrogen peroxide–induced oxidative damage, consequently acting as an antioxidant [198].

### 4.26. β-Sitosterol

β-sitosterol, relative to the central nervous system, can cross the blood–brain barrier. Hippocampus cell-line studies have reported the role of β-sitosterol derived from *Rhinacanthus nasutus* in shielding cells from neurotoxicity and oxidative damage [199]. Furthermore, the prevention of neuron damage has also been identified in a couple of studies, including inhibitory roles against the release of Aβ, β-secretase, and γ-secretase [117,118]. Stem-cell multiplication in neurons has been credited to the extract form or, otherwise, of β-sitosterol, based on in vitro and in vivo investigations. A typical finding was, for instance, identified in seeds of *Alyssum homolocarpum* [200,201].

### 4.27. Sesamin

Neuroprotectively, sesamin attenuated excess buildup of detrimental nitric oxide in the primary cells of the microglia [202]. In addition, in vivo investigation revealed neuroprotective action of sesamin in a cerebral ischemic experimental model. Furthermore, consequently, sesamin treatment, along with sesamolin, shrank the infarction region by up to 56% [202]. Sesamin pretreatment (50 µM) inhibited ROS generation intracellularly and vastly lowered proapoptotic protein expression, cellular apoptosis, and MMP-9 overexpression in SH-SY5Y cells [121].

The ameliorative effects of sesamin extract (30 mg/kg) against epileptics’ oxidative stress via the biomechanistic inhibition of MAPK and COX-2 have been reported [120]. More recently, neurite growth of PC12 cells was reportedly promoted by sesamin (10 mM), at a rate of 61.65% to 82% [122]. In this process, sesamin modulated the ERK1/2 passage, as well as the protein SIRT1 [122].

### 4.28. Myricetin

Myricetin regulates the signaling pathway BDNF–Akt/GSK-3β/mTOR, thereby enabling the regeneration of the peripheral nerve in a simulated sciatic nerve injury [124]. Notably, the nerve-regeneration-signal-related factor PI3K/Akt/mTORC1 is upregulated by myricetin (25 to 100 mg/kg). Another experimental outcome indicated that the myricetin compound, at 300 mM, reduced beta-amyloid-related cellular injury in cortical neurons [203]. Furthermore, the anti-tau effect of myricetin (50 µM) in the HeLa-C_3_ cells was elucidated [204].

### 4.29. Catechin

The anti-inflammatory and antioxidant properties contribute to the neuroprotective features of *Camellia sinensis* catechin extracts against the Parkinson’s disease model 6-OHDA [205]. Furthermore, the catechin extracts (10 mg/kg) have been reported to increase locomotor activity, reverse behavioral irregularity, and improve cognitive functionality improvement in the hippocampus [205].

Active principles and natural products can exhibit biomechanistic actions that contribute to several bioactivities related to the neuroprotective mechanism [206,207]. In this regard, Table 2 depicts some important active principles identified, along with their series of bioactive mechanisms.

## 5. Concluding Remarks

The advent of ultra-cutting edge spectroscopy in the form of the HPLC, NMR, HNMR, and FTIR will continue to remain at the heart of bioactive natural product testing. The optimization of measurability methods for natural product bioactivity, such as the in silico, in vitro, and in vivo models, as well as the cell-based and cell-line assays, will be key in driving the replicability of data neuroprotective-related research frontiers.

Furthermore, the in silico computational model will help to potentially generate the biomechanistic and chemical information of natural products. In addition, the in silico and in vitro to in vivo frontiers offer more possibilities in bioactivity response simulation.

Substantial in vitro, in vivo, and ex vivo evidence has been drawn about the capacity of naturally sourced bioactive principles (nutritive factors and secondary metabolites) to hijack or, even better, prevent the degenerative mechanisms that result in the widely known neurological diseases. Key neurologically relevant biochemical mechanisms, such as protein expression, up/downregulation, and gene targeting, provide insight into the modulatory fingerprints of these active molecules against neuropathogenesis.

In this review, several active molecules of banana and plantain fruits were identified in the peel compartment, thus offering the peel as a complementary functional food option alongside the traditionally and commonly eaten fruit pulp. Furthermore, the natural synergistic combinations among various molecules in fruits offer a bioactivity-initiation tendency. These fiber-rich peels, upon boiling, have potential dietary incorporation similar to a “vegetable”, as well as in combination with cooked vegetable, tomato, and pepper diets.

With several yet-to-be neuroprotectively implicated bioactive compounds, such as manool, palustrol, piperitenone, β-phellandrene, and cyclododecane, still in the offing, banana and plantain fruits, being biological stores of these molecules, consequently offer more potential significance in a broader pharmacological sense. Thus, the peel compartment of these fruits presents an interesting functional food option in human nutrition, as well as in further explorable neuropharmacological frontiers.

## Figures and Tables

**Figure 1 foods-11-02263-f001:**
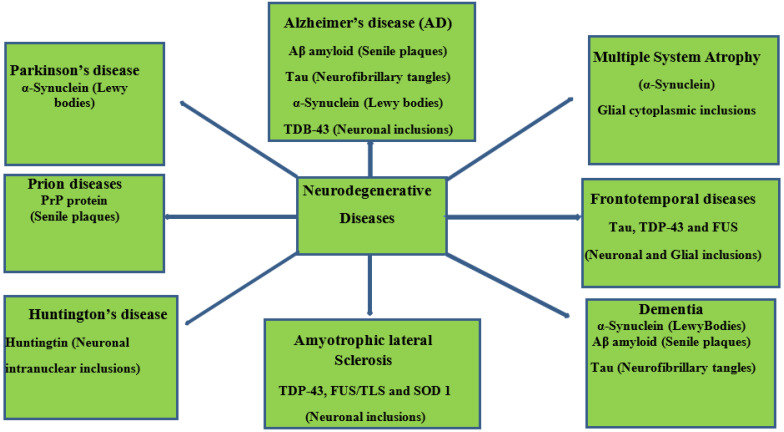
Schematic representation of neurodegenerative diseases, their pathology, and target proteins.

**Figure 2 foods-11-02263-f002:**
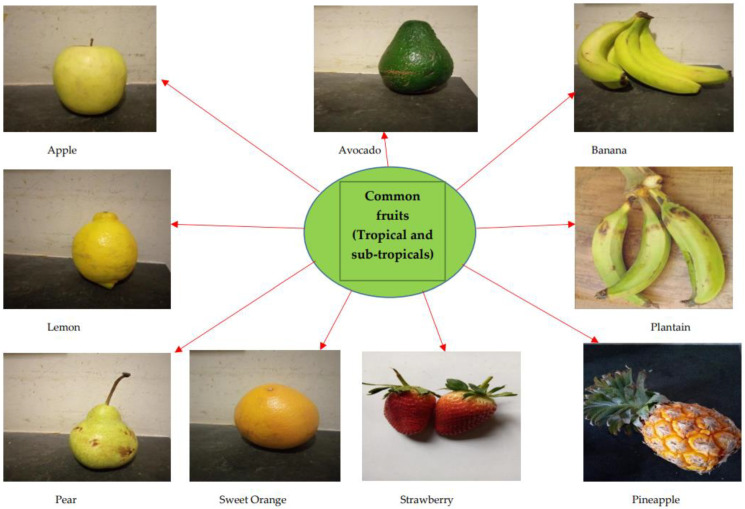
Commonly consumed fruits with bioactive capacities; including Banana and Plantain.

**Table 1 foods-11-02263-t001:** Compendium of active molecules in banana and plantain fruits.

Active Molecules (Nutritive Factors and Secondary Metabolites)	Active Molecule Constituents in Banana and Plantain FruitComponents (Pulp and Peel)	Neuromechanism-RelatedProtein/Gene Targets
**Tannin** 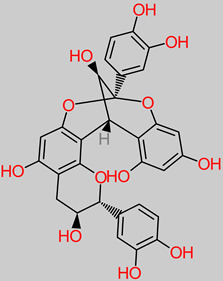	-*M. paradisiaca* peel (unripe) (5.39 ± 0.02 mg/g) [42];-*M. paradisiaca* peel (ripe) (4.24 ± 0.01 mg/g) [42];-*M. paradisiaca* peel (over-ripe) (2.84 ± 0.03 mg/g) [42];-*M. paradisiaca* fruit (2.30 ± 0.215%) [43];-*M. paradisiaca* peel extract (aqueous) (14.69 ± 0.34 mg/g) [44];-*M. paradisiaca* peel extract (80% ethanol) (17.66 ± 0.34 mg/g) [44];-*M. paradisiaca* peel extract (80% methanol);-(24.21 ± 0.17 mg/g) [44];-*M. paradisiaca* peel extract (80% acetone) (15.90 ± 0.28 mg/g) [44];-*M. paradisiaca* peel flour (30.98 ± 1.01 mgGAE/g) [45];-Cavendish banana peel (5.60 ± 0.02 mg/g) [46];-Red banana peel (5.75 ± 0.03 mg/g) [46];-White banana peel (5.00 ± 0.37 mg/g) [46];-*M. paradisiaca* raw pulp (7.05 ± 1.00 µgCE/mg) [47].	-GFAP [48];-PGC-1α/Nrf-2/Ho-1 [49].
**Phenolic acid** 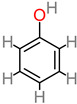	-Banana pulp (76.3 mgGAE/g) [50];-*M. acuminata* pulp (methanol extract) (778 mgQE/g) [51];-*M. acuminata* peel (methanol extract) (1168 mgQE/g) [51];-*M. acuminata* pulp (ethanol extract) (950 mgQE/g [51];-*M. acuminata* peel (ethanol extract) (897 mgQE/g) [51];-*M. paradisiaca* pulp (methanol extract) (936 mgQE/g) [51];-*M. paradisiaca* peel (methanol extract) (1346 mgQE/g) [51];-*M. paradisiaca* pulp (ethanol extract) (950 mgQE/g) [51];-*M. paradisiaca* peel (ethanol extract) (952 mgQE/g) [51];-Banana pulp extracts (150.13 to 386.22 mgGAE/100 g) [52];-*M. paradisiaca* pulp extract (166.90 ± 0.96 to 341.00 ± 34.6 mgGAE/g) [53]-*M. acuminata* (75.01 to 685.57 mgGAE/g) [54];-*M. sinensis* pulp (43.83 ± 1.13 to 119.05 ± 5.80 mgGAE/g) [55];-*M. sinensis* peel (47.68 ± 2.14 to 157.19 ± 4.76 mgGAE/g) [55];-*M. paradisiaca* pulp (17.41 ± 0.17 to 114.80 ± 1.49 mgGAE/g) [55];-*M. paradisiaca* peel (75.14 ± 0.55 to 136.87 ± 5.69 mgGAE/g) [55];-*M. acuminata* pulp (42.85 ± 0.80 to 523.60 ± 9.05 mgGAE/100 g) [56];-*M. acuminata* peel (150.48 ± 16.17 to 199.61 ± 14.68 mgGAE/ 100 g) [56].	
**Quercetin** 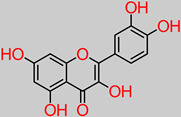	-Banana fruit (292 µg/100 g) [57].	-BDNF–TrkB-PI3K/Akt [58]-Bax/Bcl2 and caspase-3 [59].
**Rutin** 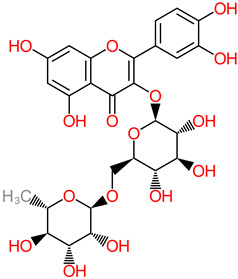	-Dessert banana (Gros Michel var.) (494.43 ± 153.71 µg/g) [60].	-Aβ, interleukin IL-1 [61,62];-TNF-α and IL-1β [63];-Poly ADP-ribosyl;-Polymerase and glutathione;-Reductase and glutathione;-Peroxidase [63];-Caspase-3 and Prion-protein peptide (Prp) [64];-Interleukin-8;-Cyclooxygenase-2, NF-kB and GFAP [65,66].-p38 MAPK [67].
**Carbohydrates** 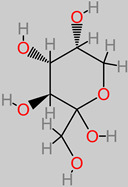	-*M. paradisiaca* pulp (18.8 to 78.5 g/100 g) [68];-*M. paradisiaca* peel (68.0 ± 0.3 g/100 g) [69];-*Musa* spp. (16.72 to 35.24 g/100 g) [70];-Banana fruit (21.70 to 41.33 g/100 g) [71].	-p-JNK and p-ERK [72];-TrKA receptor and ERK1/2;-MPTP [73].
**Lipids** 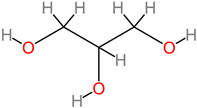	-*M. paradisiaca* (0.9 ± 0.1 g/100 g) [69];-*M. paradisiaca* (0.33 ± 0.34 g/100 g) [74].	
**Magnesium** 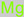	-*M. paradisiaca* peel (27.0 ± 0.08 to 76.0 ± 0.55 mg/g) [42];-*M. paradisiaca* peel flour (14.5 ± 0.0 mg/100 g) [69];-Banana peel (62.5 ± 0.01 mg/100 g) [75];-Plantain peel (64.12 ± 0.04 mg/100 g) [75];-*M. paradisiaca* pulp (29.00 ± 34.30 mg/kg) [76];-*M. paradisiaca* peel (34.50 ± 34.80 mg/kg) [76];-*M. paradisiaca* peel (324.50 ± 0.15 to 394.93 ± 0.11 mg/100 g) [77].	-N-methyl-D-aspartate [78].
**Zinc** 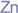	-Plantain (*Musa* ABB) and cooking banana (*Musa* AAB);-(0.2 ± 0.0 to 0.4 ± 0.0 mg/100 g) [79];-*M. paradisiaca* pulp (1.00 to 13.35 mg/kg) [76];-*M. paradisiaca* peel (3.10 to 3.70 mg/kg) [76];-*M. paradisiaca* peel (26.96 ± 0.02 to 39.02 ± 0.01 mg/100 g) [77].	-Caspase [80];-Maltose binding protein (MBP) [81].
**Copper** 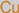	-Banana peel (2.55 ± 0.01 mg/100 g) [75];-Plantain peel (5.82 ± 0.03 mg/100 g) [75];-*M. paradisiaca* peel (3.29 ± 0.00 to 3.42 ± 0.01 mg/100 g) [77].	
**Alkaloid** 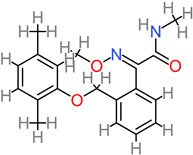	-*M. sapientum* fruit (0.251 ± 0.003 to 0.778 ± 0.006%) [43];-*M. paradisiaca* fruit (0.187 ± 0.001 to 1.027 ± 0.003%) [43];-*M. acuminata* fruit (0.083 ± 0.001 to 0.860 ± 0.005%) [43].	-Aβ peptide [82];-BDNF, MAP2, and GAP43;-PSD-95 and KLK 8 [83,84]
**Saponin** 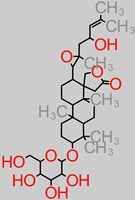	-*M. paradisiaca* peel (1.16 ± 0.82% ≅ (1160 mg/100 g) [85];-Banana fruit (11.6 mg/100 g) [71];-*M. sapientum* fruit (0.145 ± 0.005 to 2.268 ± 0.003%) [43];-*M. paradisiaca* fruit (0.773 ± 0.003 to 0.973 ± 0.033%) [43];-*M. sapientum* peel (29.25 ± 0.11 mg/100 g) [86].	-p53 and p-p38 [87].
**Phytate (Phytic acid)** 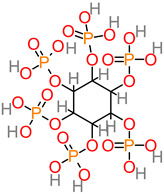	-*Musa* spp. (11.96 ± 1.05 to 24.15 ± 0.95 mg/100 g) [88];-*M. paradisiaca* peel (9.064 ± 0.04 to 11.12 ± 0.05 mg/g) [42].	-Aβ peptide [89].
**Vitamins B** 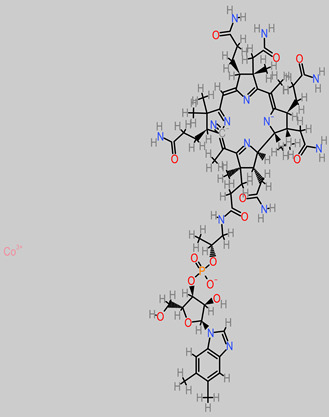	-*Musa* ABB (0.002 to 0.032 mg/100 g) [88];-*M. sapientum* pulp (0.06 ± 0.002 to 0.08 ± 0.001 mg/100 g) [43];-*M. paradisiaca* pulp (0.07 ± 0.000 to 0.08 ± 0.001 mg/100 g) [43];-*M. acuminata* pulp (0.07 ± 0.001 to 0.08 ± 0.002 mg/100 g) [43];-*M. sapientum* (0.29 ± 0.008 to 0.32 ± 0.008 mg/100 g) [43];-*M. paradisiaca* (0.24 ± 0.008 to 0.28 ± 0.118 mg/100 g) [43];-*M. acuminata* (0.30 ± 0.008 to 0.35 ± 0.012 mg/100 g) [43];-*M. paradisiaca* fruit (0.39 ± 0.02 mg/100 g) [90].	
**Anthocyanin** 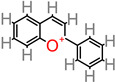	-*M. paradisiaca* peel (aqueous extract) [91].	-ASK1-JNK/p38 [92];-Interleukin-1β and TNF-α [93,94].
**Vitamin E (Tocopherol)** 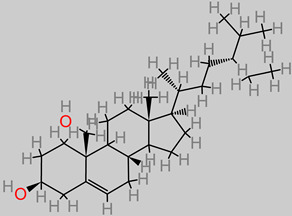	-*M. sapientum* pulp (17.53 ± 1.18 µg/g) [95];-*M. paradisiaca* pulp (20.20 ± 1.99 µg/g) [95].	-ERβ-PI3K/Akt [96].
**Selenium** 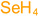	-Bananas (0.024 ± 0.0019 µg/g) [97];-Banana fruit (< 0.001 µg/g) [98];-Bananas (160 ± 1.33 µg/kg) [99];-Bananas (2.3 ± 0.20 µg/g) [100].	-Aβ peptide [101];-N-acetylcysteine [102].
**Phytosterol** 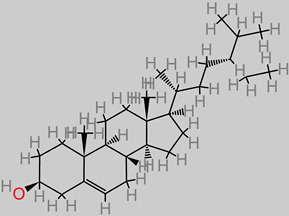	-Bananas (7.8 ± 6.9 mg/d) [103].	-Acetylcholinesterase [104];-β- secretase [105].
**Terpenoids** 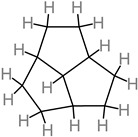	-*M. acuminata* peel (0.21 ± 0.00 to 0.28 ± 0.01 mg/g) [46];-*M. paradisiaca* peel (1.83 ± 0.19 to 1.88 ± 0.24 g/100 g) [106].	-Glutamate decarboxylase [107].
**Glycosides** 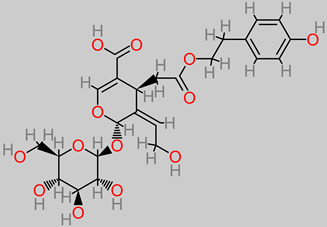	-*M. sapientum* (0.261 ± 0.001 to 0.769 ± 0.002 mg/100 g) [43];-*M. paradisiaca* (0.35 ± 0.001 to 0.602 ± 0.004 mg/100 g) [43];-*M. acuminata* (0.498 ± 0.003 to 0.811 ± 0.004 mg/100 g) [43].	-NF-kB and STAT 3 gene [108].
**Anthraquinone** 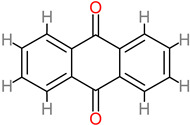	-*M. paradisiaca* peel (aqueous extract) [91].	-NF-kB and TGF-β1 [109];-ERK/MMP-9 and NOX2 (gp91phox) [110].
**Arginine** 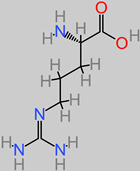	-Sweet banana fruit pulp (57.0 mg/100 g) [111].	-HIF-1α/LDHA [112].
**β-carotene** 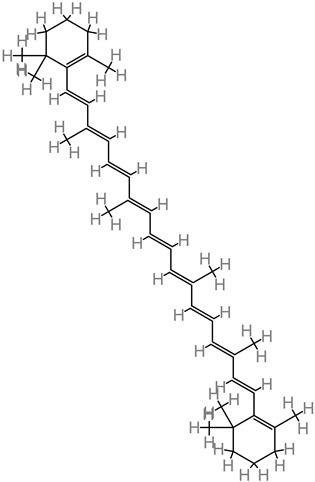	-Banana fruit (68.0 µg/100 g) [111].-Plantain fruit (390–1035 µg/100 g) [111].	-Nrf2/Keap 1 [81].
**Lycopene** 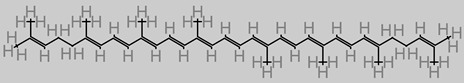	-*M. sapientum* pulp (0.80 ± 0.01 µg/g) [95];-*M. paradisiaca* pulp (0.91 ± 0.00 µg/g) [95].	-NF-kB [113];-Tau protein and GSH-Px [114].
**β-cryptoxanthin** 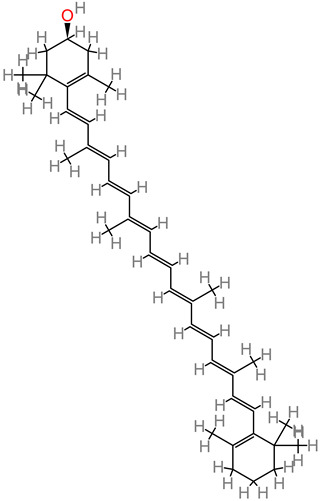	-Banana (*Musa* sp.) peel (0.08 ± 1.28 µg/g) [115].	
**β-sitosterol** 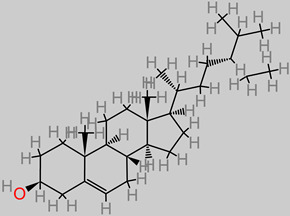	-Bananas (7.8 ± 6.9 mg/d) [103];-Banana (*Musa* sp.) peel extract (269–601 mg/kg) [116].	-Aβ, β-secretase, and γ-secretase [117,118].
**Sesamin** 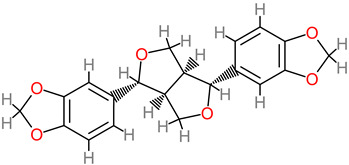	- *M. sapientum*/*M. acuminata* peel extracts (methanol) [119].	-MAPK and COX-2 [120];-MMP-9 [121];-ERK1/2 and SIRT1 [122].
**Myricetin** 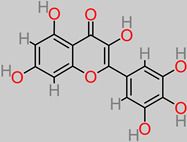	-Dessert banana peel (Grand Nain cultivar);-(125.32 ± 17.18 to 172.28 ± 12.38 µg/g) [60].-*Musa* sp. (banana fruit) (143 µg/100 g) [123].	-BDNF–Akt/GSK-3β/MTOR and P13K/Akt/MTORC1 [124].
**Catechin** 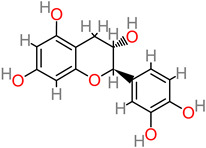	-*M. Cavendish* peel (1.34 ± 0.27% of 29.2 mgGAE/g);-Phenolic compounds [125].	
**Vitamin C (ascorbic)** 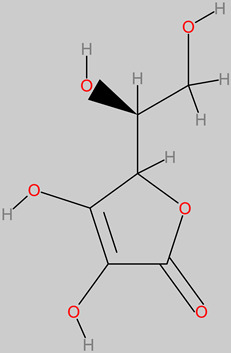	-Dessert banana (*Musa* sp.) fruit pulp (4.5 to 12.7 mg/100 g) [126].	

**Table 2 foods-11-02263-t002:** Other neuroprotective-related bioactivity mechanisms of key bioactive compounds.

Bioactive Compounds	Biological Mechanisms of Action Related to Neuroactivity	Bioactive Compounds	Biological Mechanisms of Action Related to Neuroactivity
TANNINS	- Condensed tannins and hydrolysable tannins Antioxidant (free-radical scavenging) (metal chelation) (pro-oxidative enzyme inhibition) (endogenous antioxidant system interaction) (inhibition of xanthine oxidase-induced Lipid peroxidation) [208,209].	ANTHOCYANIN	Delphinidin Cyanidin Anti-inflammatory, anti-Alzheimer, antitumor, and antioxidative properties by depleting the expression of cytokine markers [211].
	- Procyanidins Antioxidant enzyme gene expression in cells [210]. - Ellagitannins (geranin, corilagin, and furosin) Anti-inflammatory mechanisms, such as depletion of apoptotic cells [208].		
QUERCETIN	- Oxygen radical scavenging, metal chelation, and attenuation of nitric oxide synthase [212]. - Expressive mechanism of paraoxonase 2 for neuroprotection in neurons and brain cells [213,214,215]. - Anti-inflammatory mechanism via inflammatory gene repression (blocking) [216]. - Regulation of apoptosis and inhibition of cleaving enzyme (BACE 1) [217,218]. - Impairment of chemokines and cytokines [219].	MYRICETIN	Kaempferol Antitumor, anti-inflammation, and antioxidant properties are exercised via an antiproliferative mechanism in cells, attenuation mechanism against inflammation, and tumor growth factors [124,221].
	- Quercetin-3-*O*-diglucoside-7-*O*-glucoside Anti-inflammatory, antioxidant effects, and lipoxygenase inhibitory effects [220].		
ALKALOIDS	Vincristine Antineuroblastoma property exerted via the mechanisms truncating the glutathione metabolism [222]. Tetrandrine Anti-inflammatory and antitumor activities are linked to the calcium-channel blocking mechanism [223]. Skimmianine Anti-inflammatory property via the inhibition of nitric oxide production [224].	CATECHIN	- Anti-inflammation and antioxidative stress mechanism via modulation of tyrosine kinase receptor [225]. - Modulation of signal transduction pathways to protect cell proliferation, inflammation, and metastasis [226].
TERPENOIDS Paeoniflorin, Triptolidenol, Tripterine, Triptonide, Gindenoside, Oleanoic Acid	Anti-inflammatory activity via interleukin-6 inhibition [227]. Anti-nociceptive, antioxidant and anti-inflammatory properties [228].	RUTIN	- Anti-apoptotic mechanism against cell death [112]. - Depletion of pro-inflammatory cytokine expression [229].
LIPID Omega-3 DHA	Anti-inflammatory properties, Cell survival promotors [230].	Neuroprotectin D1 Lipid	Inhibition of apoptosis-related damage to DNA [230].

## Data Availability

Data are available in the open-access article.

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
