# Peer review of "Suitability of Banana and Plantain Fruits in Modulating Neurodegenerative Diseases: Implicating the In Vitro and In Vivo Evidence from Neuroactive Narratives of Constituent Biomolecules"

_foods, 2022, doi:10.3390/foods11152263_

Round 1

Reviewer 1 Report

This is a very well-written review paper.

Some minor points for further improvement:

  1. Figure 1 needs editing, there are some arrows out of place,
  2. Table 1: all atoms need to be in black (not red),
  3. section 4.6. on Lipids: the authors need to name which omega-3 lipid refer to. Also, I would invite the authors to add here information on polar lipids and their anti-inflammatory properties, hence polar lipids should be mentioned here (e.g. https://www.sciencedirect.com/science/article/pii/S2352939318300010)
  4. in the section 5. concluding remarks, I would invite the authors to provide their views on possible future research themes that would generate solid data on these bioactivities.

Author Response

We would like to submit the authors’ responses to the comments of Reviewer 1 for the manuscript (foods-1495210) titled: “Suitability of Banana and Plantain fruits in modulating neurodegenerative diseases: Implicating the in vitro, in vivo evidences from neuroactive narratives of constituent biomolecules.”

1. Figure 1 needs editing, there are some arrows out of place.

Response: Figure 1 has been revised appropriately. Alterations identified may have been due to some editorial formatting effects.

2. Table 1: all atoms need to be in black (not red)

Response: Atoms are generated from the Chemistry database software and this makes it quite difficult to colour-edit these atoms. The action appears to be locked in the software.

3. Section 4.6. on Lipids: the authors need to name which omega-3 lipid refer to. Also, I would invite the authors to add here information on polar lipids and their antiinflammatory properties, hence polar lipids should be mentioned here (e.g. 
https://www.sciencedirect.com/science/article/pii/S2352939318300010)

Response: The omega-3 lipid referred to is Docosahexaenoic acid (DHA). Furthermore, polar lipids (fatty acyls) such as the Omega-3 polyunsaturated fatty acid, as well as its derivative bioactive lipid Neuroprotection D1 (NPD1) possess distinct antiinflammatory properties, are promotors of cell survival and neuroregeneration in the cornea, and are inhibitors of apoptosis-related DNA damage (Shammim et al, 2018). 
This information has been added to the revised manuscript reflecting mechanism actions of bioactive compounds.

4. In the section 5. Concluding remarks, I would invite the authors to provide their views on possible future research themes that would generate solid data on these bioactivities.

Response: The advent of ultra-cutting edge spectroscopy in form of the HPLC, NMR, HNMR, and FT-IR will continue to remain at the heart of bioactive natural product testing. The optimization of measurability methods for natural product bioactivity, such as the in silico, in vitro, in vivo models, cell-based and cell-line assays, will be key in driving the replicability of data neuroprotective-related research frontiers. 
Furthermore, the in silico computational model will help to potentially generate 
biomechanistic and chemical information of natural products. In addition, the in silico and in vitro to in vivo frontiers, offer more possibilities in bioactivity response simulation.

Reviewer 2 Report

Interesting review however the authors should correct appropriately.

  1. Fig. 1 needs to make a better presentation, the arrows are everywhere.
  2. Table 1 please clear out what is in the parenthesis, you refer to Banana and Plantain fruit compartmental constitution.
  3. what is GAE and what is QE?
  4. This table only compares 2 parameters for active molecules and is too long to read.
  5. Authors should use additional tables to illustrate any differences between other parameters too. Additional figures should also be used.

Author Response

We would like to submit the authors’ responses to the comments of Reviewer 2 for the manuscript (foods-1495210) titled: “Suitability of Banana and Plantain fruits in modulating neurodegenerative diseases: Implicating the in vitro, in vivo evidences from neuroactive narratives of constituent biomolecules.”
1. Fig. 1 needs to make a better presentation, the arrows are everywhere.
Response: Figure 1 has been revised appropriately. Alterations may have been due to some editorial formatting.

2. Table 1 please clear out what is in the parenthesis, you refer to Banana and Plantain fruit compartmental constitution.
Response: In Table 1, the column “Banana and Plantain fruit compartmental 
constitution” has been reworded to read “Active molecule constituents in Banana and Plantain fruit compartments”.

3. What is GAE and what is QE?
Response: GAE refers to “Gallic Acid Equivalent”, while QE refers to “Quercetin 
Equivalent”.

4. This table only compares 2 parameters for active molecules and is too long to read.
Response: Table 1 compares the active molecules of the pulp and peel of banana and plantain, including their sub-components such as the peel extract and pulp extracts.
Table 1 appears to be long, particularly because of the effect of the chemical structures of the molecules which are software-generated. Thus, resizing these structures is proving to be quite difficult.

5. Authors should use additional tables to illustrate any differences between other parameters too. Additional figures should also be used.
Response: Figure 2 has been inserted to depict other fruits that have been identified with bioactivity, along with banana and plantain. A number of these fruits also have the related bioactive functions as neuroprotective, anti-inflammatory and antioxidant properties

Reviewer 3 Report

The paper entitled ‘Suitability of Banana and Plantain fruits in modulating neurodegenerative diseases: Implicating the in vitro, in vivo evidence from neuroactive narratives of constituent biomolecules’ is aimed to discuss the influence of selected natural compounds and microelements on neurodegeneration . The subject of the paper is interesting nevertheless in my opinion, the issue is presented superficial.

Authors present activity of selected group of compounds (i.e. tanins, terpenoids) but the presentation does not include detailed information. In all cases, authors include information about activity of selected compounds and/or possible mechanism of action. In accordance with my knowledge, the presented compounds can act with various mechanisms which should be presented. One of a few possible mechanisms is insufficient to present concrete compound as possible natural substance counteracting dementia. Additionally, Fig 1. Must be improved.

I recommend enrich descriptions of all compounds with other possible mechanisms of action or present it in form of scheme.

Author Response

We would like to submit the authors’ responses to the comments of Reviewer 3 for the manuscript (foods-1495210) titled: “Suitability of Banana and Plantain fruits in modulating neurodegenerative diseases: Implicating the in vitro, in vivo evidences from neuroactive narratives of constituent biomolecules.”
Authors present activity of selected group of compounds (i.e. tanins, terpenoids) but the presentation does not include detailed information. In all cases, authors include information about activity of selected compounds and/or possible mechanism of action. In accordance with my knowledge, the presented compounds can act with various mechanisms which should be presented. One of a few possible mechanisms is insufficient to present concrete compound as possible natural substance counteracting dementia. Additionally, Fig 1. Must be improved.
I recommend enrich descriptions of all compounds with other possible mechanisms of action or present it in form of scheme.

Response: Table 2 has been constructed to present some key bioactive compounds, their subtypes and mechanisms of action in bioactivities which are relatively linked to neuroprotective tendencies in the body.
In addition, figure 1 has been revised appropriately (alterations observed may have been due to the editorial formatting effects)

Reviewer 4 Report

The manuscript  entitled “Suitability of Banana and Plantain fruits in modulating neurodegenerative diseases: Implicating the in vitro, in vivo evidence from neuroactive narratives of constituent biomolecules” describes plant-based foods especially staple fruits like banana and plantain and their inter-related anti-inflammatory, anti-apoptotic, antioxidative and, neuromodulatory activities. The manuscript is very interesting and the experiments are generally well planned and described. However, Authors should correct manuscript according to the suggestion.                                                                                     

Minor issues:

  • Table 1 please explain abbreviations: GAE, QE and EC
  • I propose separate section for minerals and their properties.

Author Response

We would like to submit the authors’ responses to the comments of Reviewer 4 for the manuscript (foods-1495210) titled: “Suitability of Banana and Plantain fruits in modulating neurodegenerative diseases: Implicating the in vitro, in vivo evidences from neuroactive narratives of constituent biomolecules.”
Table 1 please explain abbreviations: GAE, QE and EC. I propose separate section for minerals and their properties.

Response: GAE refers to “Gallic Acid Equivalent” QE refers to “Quercetin Equivalent” EC has been corrected to read CE, which refers to “Catechin Equivalent”.
The section neuroprotective mechanistic narratives of active principles and mineral elements listed in Banana and Plantain fruits has been designed to accommodate biological properties of some mineral elements.

Round 2

Reviewer 2 Report

Authors have revised sufficiently and paper can be accepted

Reviewer 3 Report

The manuscript has been corrected in accordance with suggestions. It can be accepted in present form.